# A systematic review of the limitations of large language models in generating healthcare content

Mohsen Khosravi[1]*, Zahra Zamaninasab[2], Seyyed Morteza Mojtabaeian[3], Emine Kübra Dindar Demiray[4], Morteza Arab-Zozani[1]

**1** Social Determinants of Health Research Center, Birjand University of Medical Sciences, Birjand, Iran, **2** Department of Epidemiology and Biostatistics, School of Health, Social Determinants of Health Research Center, Birjand University of Medical Sciences, Birjand, Iran, **3** Department of Healthcare Services Management, School of Management and Medical Informatics, Shiraz University of Medical Sciences, Shiraz, Iran, **4** Department of Infection Diseases and Clinical Microbiology, Siirt University Medical School, Siirt, Türkiye

* mohsenkhosravi@live.com

## Abstract

Large language models (LLMs) have recently gained prominence in healthcare content provision due to their numerous advantages. Despite these benefits, LLMs exhibit notable limitations in this domain. This study aimed to systematically identify the limitations of LLMs in provision of healthcare content. This study was a systematic review conducted in September 2025, including articles published in English between 2018 and 2025. Searches were performed in PubMed, Scopus, and the Cochrane Database of Systematic Reviews. Two independent evaluators screened the references and assessed quality of the selected studies using the Authority, Accuracy, Coverage, Objectivity, Date, and Significance (AACODS) checklist. Data were analyzed using Boyatzis's qualitative thematic approach with an inductive methodology, applying the input–process–output (IPO) model as the analytical framework. A total of 81 studies were included in the final analysis. The included studies were predominantly of high quality and demonstrated minimal risk of bias. The thematic analysis identified key themes: data limitations, dependence on input and prompt quality, accessibility issues, model design and architecture constraints, interaction challenges, response quality and comprehensiveness, and ethical, safety, and regulatory concerns. The study identified multiple limitations of LLMs in healthcare, with output issues being most common. In this regard, the most frequently cited limitation was the accuracy gap. However, these output issues were mainly resulted from flaws in input data, emphasizing the crucial role of input quality. The study also proposed strategies to address these challenges.

**Data availability statement:** The research data is presented as a supplementary file.

**Funding:** The author(s) received no specific funding for this work.

**Competing interests:** The authors have declared that no competing interests exist.

## Author summary

This study systematically reviewed the existing literature regarding the limitations of large language models(LLMs) in provision of healthcare content. A group of prominent databases were searched and the screened references were assessed in terms of quality. Finally, a thematic analysis was conducted on the data derived from the included studies corresponding to the research question using the input-output model as an analytical framework. The search yielded 81 studies and the quality of the included studies was presented to be predominantly high. The thematic analysis yielded a number of themes and sub-themes. The results of the research found that while the main area of LLMs` limitations is corresponding to the outputs, and particularly the existing gaps in accuracy, these limitations are shown to be derived from the existing flaws in the input data. The study also presented some strategies to overcome these limitations based on the existing data within the literature.

## 1. Introduction

Large language models (LLMs) are sophisticated artificial intelligence (AI) systems developed through extensive training on vast corpora of text data, enabling them to generate outputs that closely resemble human language [1]. These models have been widely utilized across diverse medical domains, including health informatics, medical imaging, clinical diagnostics, treatment planning, ophthalmology, oncology, and other specialized fields [2]. This trend signifies their broad and growing integration into medical research and clinical practice.

LLMs have become pivotal in healthcare by enhancing clinical decision support, diagnostics, medical education, and patient engagement [2–4]. They improve diagnostic accuracy by analyzing extensive clinical data and medical literature, aiding in personalized treatment planning and patient care management [3]. LLMs are increasingly integrated into hospitals, clinical settings, academic medical centers, and virtual care platforms, supporting healthcare providers with evidence-based recommendations and facilitating patient interactions through chatbots and virtual assistants [2]. Furthermore, they assist in research by automating documentation and synthesizing biomedical information, thereby optimizing clinical workflows [5].

Notwithstanding the numerous advantages previously discussed, LLMs in healthcare exhibit several critical limitations that must be addressed to ensure their safe and effective deployment. In this regard, they are prone to generating plausible yet factually incorrect or fabricated information, a phenomenon known as hallucination, which poses significant risks in clinical settings [3,6]. Furthermore, LLMs often lack the depth of contextual understanding required to accurately interpret complex medical scenarios, as they may fail to integrate multifaceted clinical data or temporal information adequately [3]. The development and application of LLMs are also constrained by limited access to high-quality, diverse clinical datasets due to privacy,

ethical, and legal challenges [7]. Ethical and legal issues such as bias, misinformation, data privacy, and insufficient regulatory frameworks further challenge their adoption [8]. Additionally, the opaque, "black box" nature of these models undermines transparency and interpretability, complicating trust and reliance by healthcare professionals [9]. Practical concerns include their substantial computational and energy demands, which limit feasibility in resource-constrained environments [3]. Such limitations of LLMs pose significant challenges to their adoption and utilization across various sectors of healthcare systems. In this regard, these challenges must be critically addressed to enable the successful and comprehensive integration of LLM technologies within healthcare infrastructure.

While several review studies have aimed to delineate the limitations of LLMs in providing healthcare content, there remains a significant gap in the literature regarding a comprehensive and systematic presentation of these limitations for end-users [3,6–9]. In this regard, a systematic review categorized the limitations of LLMs into two primary domains: design and output. Design limitations included several items such as lack of optimization for the medical domain, data transparency issues, and accessibility challenges. Output limitations also included several items such as non-reproducibility, incompleteness, inaccuracies, safety concerns, and biases [6].

The data generated from research on the limitations of LLMs would be invaluable for technology developers to enhance the quality of their models. Additionally, healthcare policymakers and administrators could leverage this information to make informed decisions about implementing these technologies within their organizations, fully acknowledging their current constraints. Moreover, future researchers would benefit from this detailed framework by conducting focused investigations on each identified limitation, thereby contributing further insights and advancing the field for subsequent users and stakeholders.

## 2. Results

As demonstrated in Fig 1, the database search was conducted on September 13, 2025, yielding 409 references from the Cochrane Database of Systematic Reviews, 1,766 references from PubMed, and 3,278 references from Scopus, of which 1,798 were identified as duplicates. Following the screening of the retrieved references, a total of 81 studies were included as the final selections for the study. 20% of the included studies were published in 2023, 40% in 2024, and the remaining 40% in 2025.

### 2.1. Data quality

The quality assessment of the included studies indicated that they were predominantly of high quality, with an average score of 10. Approximately 34% of the studies achieved the highest quality score of 12, while about 8% scored 7, reflecting lower quality relative to the other included studies. Moreover, as delineated by the data corresponding to the objectivity item of the AACODS checklist, the level of bias in the included studies was generally low, with approximately 59% of the studies exhibiting the minimal possible bias (S1 Appendix).

### 2.2. Data analysis

The thematic analysis identified a total of eight distinct themes within the four major categories outlined by the IPO model concerning the limitations of LLMs in healthcare content generation. The themes included data limitations, dependence on input quality, dependence on prompt quality, accessibility issues, model design and architecture limitations, interaction challenges, response quality and comprehensiveness, and ethical, safety, and regulatory concerns (Table 1).

**2.2.1. Input limitations. 2.2.1.1. Data limitations:** The scarcity of medical dialogue datasets in Arabic was highlighted as a critical constraint, primarily due to privacy concerns and the sensitive nature of medical conversations, which results in limited availability of comprehensive and representative datasets [10]. Additionally, the effective functioning of these models was found to heavily depend on access to comprehensive, unbiased, and up-to-date training data. Any gaps or

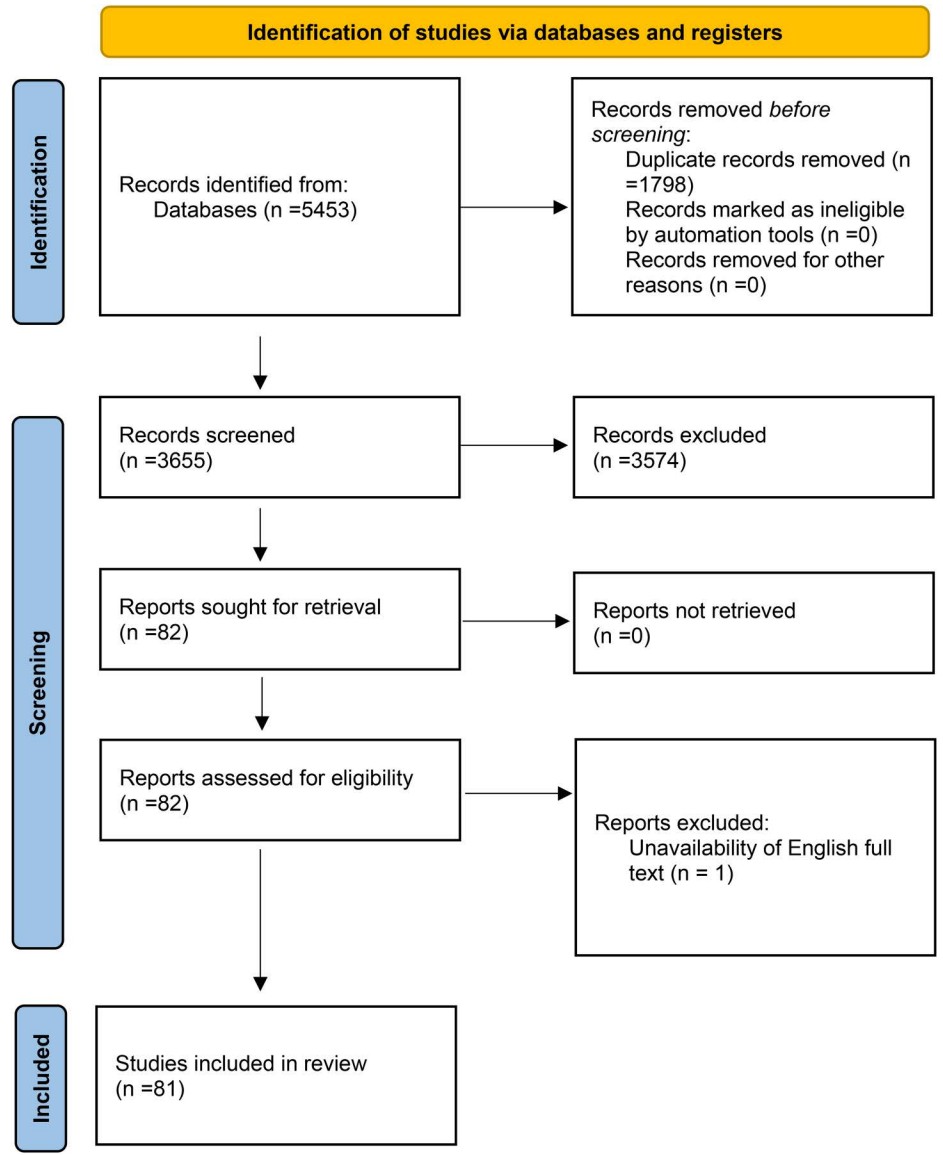

**Fig 1. PRISMA diagram.**

deficiencies in such data adversely affected the quality and reliability of the generated outputs, underscoring the essential role of robust and well-maintained datasets in ensuring model accuracy and relevance within healthcare contexts [11–17].

 **2.2.1.2. Dependence on input quality:** A key issue identified was the need for structured and clear inputs, as unstructured or ambiguous data inputs were found to reduce AI accuracy and effectiveness in clinical decision-making [18,19]. The complexity of some languages such as the Arabic language, characterized by its rich morphology and diverse dialects, further complicates natural language processing tool development compared to more standardized languages [10]. Additionally, the models exhibited a limited ability to recognize non-verbal signals, such as subtle crisis indicators or nuanced mental states, which restricts their utility in critical situations like suicidal or homicidal ideation [20].

**Table 1. Thematic analysis of findings.**

| Category | Theme | Sub-theme | Reference(s) |
|---|---|---|---|
| Input limitations | Data limitations | Scarcity of Medical Dialogue Datasets | [10] |
| | | Dependence on Comprehensive Training Data | [11–17] |
| | Dependence on input quality | Need for Structured and Clear Inputs | [18,19] |
| | | Complexity of Language and Dialects | [10] |
| | | Limited Ability to Recognize non-verbal Signals | [20] |
| | | Session and Question Limits | [21] |
| | | Limited Multimodal Data Processing | [13,19,22–25] |
| | | Issues with question difficulty and length | [26] |
| | Dependence on prompt quality | Need for Prompt Engineering | [13,24,25,27–38] |
| | | Vulnerability to Adversarial prompts | [39] |
| | Accessibility issues | Dependency on access to model internals and updates | [33,40–43] |
| | | Dependency on Stable Internet | [44] |
| Process limitations | Model Design and Architecture Limitations | High Power Consumption | [44,45] |
| | | Algorithmic Bias | [13–15,19,20,23,25,40,41,43,45–54] |
| | | Hardware Constraints | [43,44] |
| | | Lack of Clinical Experience and Judgment | [19] |
| | | Over-reliance on imaging modalities | [13,29,55] |
| | | Inability for Critical Thinking | [30,41,56] |
| | | Reliance on Provided Data | [56] |
| | Interaction Challenges | Requirement for Repeated Interactions | [16,24,57] |
| | | Lack of interaction | [23,56,58] |
| | | Conversation Tracking Issues | [21,59] |
| | | Latency Issues | [44] |
| | | Inability to Ask Clarifying Questions | [60] |
| Output limitations | Response Quality and Comprehensiveness | Limited Depth of Responses in Complex Assignments | [12,13,16,18,22–24,26–28,30,31,36,37,42,49,50,53,56,58,60–73] |
| | | Restricted Response Length | [27,61,74] |
| | | Incomplete responses | [40] |
| | | Inconsistency in Responses | [11,15,19,28,40,50,51,63,75–77] |
| | | Repetitive and Vague Recommendations | [15,30,56] |
| | | Lack of Personalization and Clinical Nuance | [17,20,23,30,35,37,45,53,57,58,60,66,68,72,73,76,78–82] |
| | | Limited Actionability | [11,14,17,25,27,31,50,55,63] |
| | | Accuracy Gaps | [12,13,15,16,19–21,23,25,26,28,29,31,34–37,40,42,43,46,48,49,51,53,55–58,64,65,68–70,72,73,75–77,79–86] |
| | | Advanced Reading Level | [15,17,25,27,31,32,34,50,67,70,81,87,88] |
| | | Variable Translation Quality by Language | [10,32,67,76,77,80,89] |
| | | Outdated References | [11,19,41,50,51,55,57,59–61,76] |

*(Continued)*

**Table 1.**  (Continued)

| Category | Theme | Sub-theme | Reference(s) |
|---|---|---|---|
| | Ethical, Safety, and Regulatory Concerns | Ethical, and Regulatory Challenges | [11,13,14,19,20,23,35,39–41,43–48,50,52–54,75,76,84,86,90,91] |
| | | Regulatory and Quality Control Issues | [46] |
| | | Risk of Overreliance | [20,22,37,52,53,65,76,78,91] |
| | | Risk of Misinterpretation | [76] |
| | | Lack of Transparency | [11,34,40,53,70] |
| | | Lack of Real-World Validation | [16,23,25,35,43,66] |
| | | Inconsistent Use of Disclaimers | [14,21,34,67] |
| | | Inability to Replace Human | [14,19,20,22,23,26,35,36,40,42,45–47,49,51,57,58,60,68,69,71,73,76,79–81,83,85,88] |
| | | Lack of Self-awareness | [65] |
| | | Risk of dangerous information | [39,84] |
| | | Risk of Misinformation in Less Common Languages | [32,67,76,77,80,89] |

Session and question limits, such as the ChatGPT 4.0 restriction of 40 questions per three hours—absent in version 3.5—also potentially hinder the model's effectiveness as a telepharmacy or healthcare tool [21]. Moreover, current large language models possess limited multimodal data processing capabilities, primarily handling text with minimal ability to interpret images or other data types, thereby constraining their applicability in fields heavily reliant on imaging, including radiology and pathology [13,19,22–25]. Finally, the complexity and length of questions were negatively correlated with accuracy, as longer and more difficult questions were more likely to be answered incorrectly [26].

**2.2.1.3. Dependence on prompt quality:** Optimized prompt engineering is essential to enhance clarity, actionability, and readability of model outputs while minimizing the risk of misinformation [13,24,25,27–38]. Additionally, these models demonstrated vulnerability to adversarial prompts, whereby maliciously crafted inputs could circumvent existing safeguards. For instance, even advanced models such as ChatGPT-4.0 were susceptible to manipulation that enabled generation of harmful and detailed instructions, including those that could potentially cause ocular damage through biological, chemical, or physical means [39].

**2.2.1.4. Accessibility issues:** Restricted access to model internals and updates was found to limit broader implementation and comprehensive understanding of these models within healthcare settings [33,40–43]. Furthermore, most large language models require a stable internet connection for cloud-based processing, which introduces latency and significantly restricts their usability in offline environments or regions with poor connectivity [44].

**2.2.2. Process limitations. 2.2.2.1. Model design and architecture limitations:** High power consumption associated with cloud-based LLM usage was identified as a constraint, particularly for portable or energy-sensitive applications [44,45]. Algorithmic bias present in training data, such as racial bias, resulted in models like ChatGPT producing varied recommendations based on patient race or ethnicity, thereby perpetuating healthcare disparities and obscuring such biases due to the models' opaque "black box" nature [13–15,19,20,23,25,40,41,43,45–54].

Hardware constraints were also noted, with limited memory and processing capacity of devices like microcontrollers (e.g., ESP32 and ESP8266) posing challenges to the complexity and responsiveness of LLM-powered healthcare applications [43,44]. Additionally, the lack of real-world clinical experience restricted LLMs' ability to perform complex clinical reasoning, diagnostic accuracy, and higher-order judgment essential for medical decision-making [19].

There was also an identified over-reliance on imaging modalities such as CT and MRI, without adequate customization for specific clinical contexts [13,29,55]. Furthermore, LLMs like ChatGPT demonstrated an inability for critical thinking

necessary to tailor and guide patient management effectively [30,41,56]. Finally, responses generated by these models depended solely on the provided data without interpretative insight or clinical opinion, risking omission of underlying clinical nuances—such as failing to detect depression in patients presenting with non-specific symptoms like sleep disturbances [56].

**2.2.2.2. Interaction challenges:** Effective use by clinicians often required repeated interactions, involving multiple queries and refined questioning to obtain accurate and relevant responses [16,24,57]. However, the models lacked the capability for dynamic interaction, as they could not gather additional information necessary for precise diagnosis and management [23,56,58].

Conversation tracking posed further limitations: ChatGPT 3.5 lacked conversation memory due to privacy and browsing restrictions, while ChatGPT 4.0, despite tracking conversations, inaccurately counted inquiries, thereby undermining feedback and dialogue continuity [21,59]. Latency issues arising from data transmission to cloud servers also presented challenges, potentially delaying real-time healthcare applications that demand immediate responses [44]. Moreover, these models were unable to ask clarifying questions to seek further clinical clues, which constrained their diagnostic accuracy and the precision of their advice [60].

**2.2.3.   Output limitations.  2.2.3.1. Response quality and comprehensiveness:** LLMs often provided limited depth in complex tasks, offering superficial or incomplete answers, particularly for clinical interventions, follow-up discussions, critical appraisals, and complex data reasoning, resulting in poorer performance compared to simpler question types [12,13,16,18,22–24,26–28,30,31,36,37,42,49,50,53,56,58,60–73]. Additionally, response length was constrained by a maximum word count (approximately 650 words for ChatGPT), limiting comprehensive critical analysis and extensive discussion of complex healthcare topics, though improvements are anticipated in newer model versions [27,61,74]. Incomplete responses were also noted, including occasional neglect of imaging descriptions [40].

Consistency issues arose as LLMs sometimes generated varying answers to identical questions or prompts, undermining reliability [11,15,19,28,40,50,51,63,75–77]. Further prompting often produced broad, repetitive, and vague recommendations lacking personalization or precision [15,30,56]. Responses frequently missed subtle clinical nuances and tailored patient-specific details, necessitating professional oversight to ensure accuracy and patient safety [17,20,23,30,35,37,45,53,57,58,60,66,68,72,73,76,78–82].

While chatbot outputs were generally understandable, they were often insufficiently actionable, limiting patients' ability to take clear steps, and despite streamlining some administrative tasks, LLMs currently offer limited impact on routine clinical care without further development [11,14,17,25,27,31,50,55,63]. Accuracy gaps were evident, with occasional imprecision, indecisiveness, hallucinations, irrelevant information, and omissions of key clinical considerations, particularly for special populations such as pregnant patients. Moreover, fabricated references were also observed sporadically [12,13,15,16,19–21,23,25,26,28,29,31,34–37,40,42,43,46,48,49,51,53,55–58,64,65,68–70,72,73,75–77,79–86].

The model's language was often at an advanced reading level, complicating accessibility and comprehension for many patients [15,17,25,27,31,32,34,50,67,70,81,87,88]. Moreover, translation quality varied across languages, and references cited were frequently outdated, with a tendency to neglect recent high-quality studies such as randomized controlled trials, reducing the reliability of evidence-based content, especially in critical appraisals where current literature is essential [10,11,19,32,41,50,51,55,57,59–61,67,76,77,80,89].

**2.2.3.2. Ethical, safety, and regulatory concerns:** Ethical and regulatory challenges emphasized the need for enforceable regulations, comprehensive ethical frameworks, and proactive controls to prevent misuse, particularly given the sensitivity of AI applications in healthcare and biowarfare. Issues of authorship, accountability, and trustworthiness were raised as AI cannot assume responsibility for its outputs and may perpetuate biases embedded in training data. Patient data privacy and vulnerability to cyber threats further necessitate robust data protection measures [11,13,14,19,20,23,35,39–41,43–48,50,52–54,75,76,84,86,90,91].

Regulatory and quality control challenges were highlighted due to the evolving nature of AI models and the current underdevelopment of regulatory frameworks governing algorithmic medicine [46]. The risk of overreliance emerged as a critical concern, with users potentially accepting AI-generated information without critical evaluation or expert consultation, posing risks of misinformation and harm. Excessive dependence on ChatGPT was also associated with increased social isolation, potentially exacerbating depression in vulnerable populations [20,22,37,52,53,65,76,78,91].

Misinterpretation risks were identified due to variability in user comprehension, which could lead to adverse clinical outcomes [76]. The lack of transparency stemming from the model's "black box" operation limits the ability to evaluate responses, as the rationale behind conclusions remains unclear [11,34,40,53,70]. Additionally, the absence of real-world clinical validation restricts independent clinical use [16,23,25,35,43,66]. The inconsistent use of disclaimers—more frequent in earlier versions like ChatGPT 3.5, but less so in newer versions—raises questions regarding role compliance [14,21,34,67].

AI's inability to replace human judgment was noted, given its lack of genuine understanding, empathy, and ethical reasoning, functioning instead as a "stochastic parrot" mimicking language without true comprehension [14,19,20,22,23,26,35,36,40,42,45–47,49,51,57,58,60,68,69,71,73,76,79–81,83,85,88]. The model demonstrated limited self-awareness, seldom acknowledging when it lacked answers and frequently providing incorrect explanations without recognizing these limitations [65]. Concerns about the risk of generating dangerous information were amplified by potential exploitation by malicious actors [39,84]. Finally, the variability and higher error rates in translations for less common languages heighten the risk of miscommunication in clinical contexts when relying solely on machine-generated translations [32,67,76,77,80,89].

## 3. Discussion

As presented by the study findings and delineated in Fig 2, which presents the number of citations for each sub-theme presented in the thematic analysis, the category with the greatest number of limitations identified in the literature was output limitations, underscoring the significant challenges related to the output of large language models in the provision of healthcare content. In line with our study findings, a recent review emphasized that the taxonomy of limitations associated with large language models reveals a substantially greater number of codes related to output issues compared to those connected with design or input phases. These output limitations encompass challenges such as generating accurate, contextually relevant, and comprehensive responses, as well as concerns regarding interpretability, responsiveness, and ethical considerations [3].

The findings of our study identified accuracy gaps (i.e., fabricated responses) as the most frequently reported limitation of LLMs. This provides the rationale for the notable increase in recent studies evaluating the accuracy of LLMs [92,93]. In accordance with these findings, which highlight the significance of limitations related to the output of LLMs, this section is primarily dedicated to analyzing the existing data on this subject, taking into account findings from the literature in other contexts.

The findings of our study identified several output limitations of LLMs in generating healthcare content. These limitations included issues with response quality and comprehensiveness, such as limited depth of responses in complex assignments, restricted response length, incomplete answers, inconsistency, as well as repetitive and vague recommendations. Additionally, the models demonstrated a lack of personalization and clinical nuance, limited actionability, and accuracy gaps (i.e., fabricated responses). Other challenges involved advanced reading levels, variable translation quality across languages, and reliance on outdated references. Ethical, safety, and regulatory concerns were also evident, including regulatory and quality control issues, risks of overreliance and misinterpretation, as well as lack of transparency and real-world validation. The inconsistent use of disclaimers, inability to replace human expertise, lack of self-awareness, and risks of disseminating dangerous information or misinformation—particularly in less common languages—were further significant limitations noted [10–32,34–37,39–91].

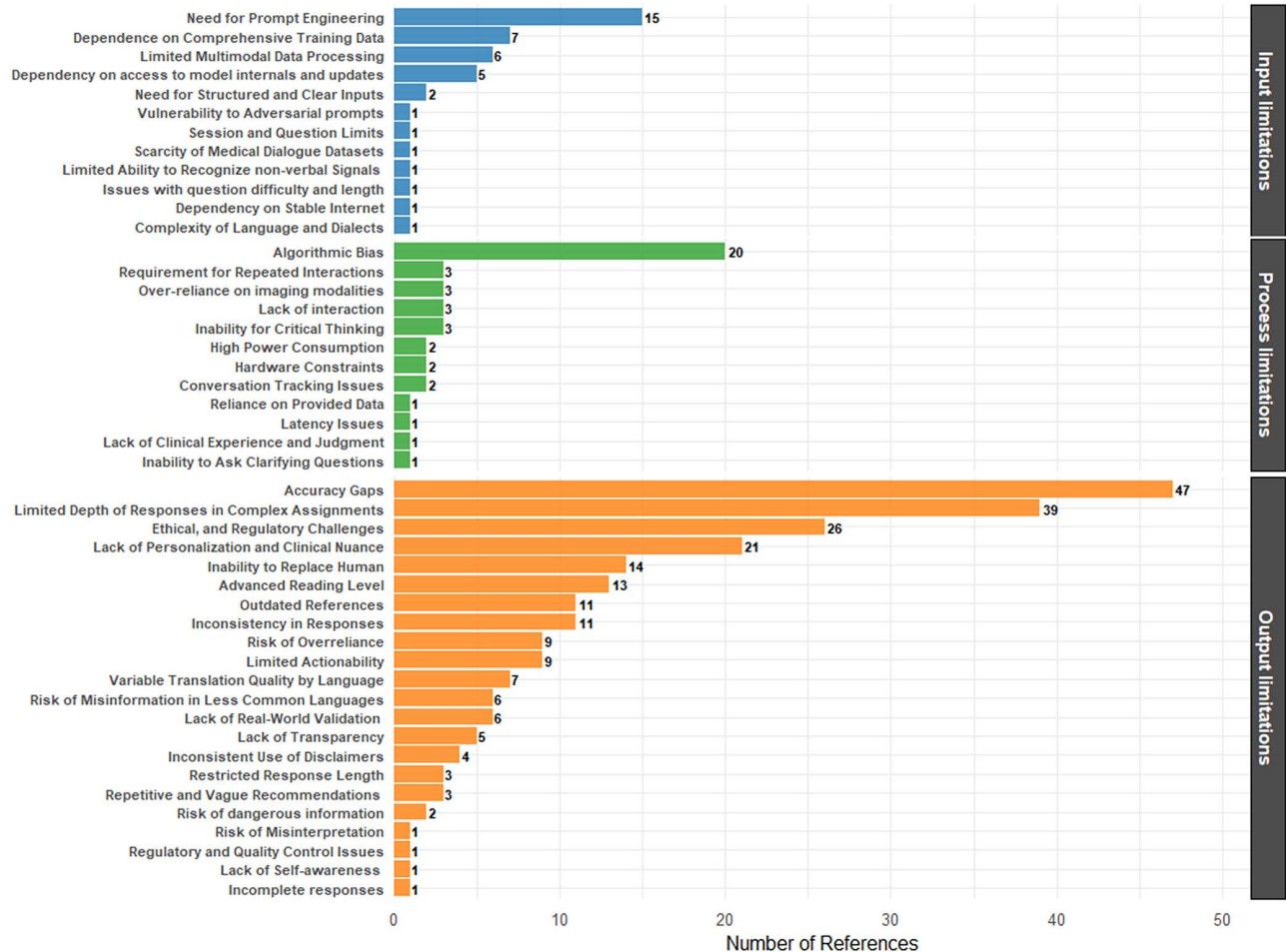

**Fig 2. Distributions of limitations of LLMs, categorized according to the frequency of citations reported in the literature.**

The literature has identified multiple reasons for the issues related to response quality and comprehensiveness in LLMs. In this regard, a significant portion of these issues originates from limitations in the training data. For instance, LLMs frequently generate incomplete answers due to gaps in their training datasets and their inability to fully integrate complex or comprehensive information. This shortcoming is particularly critical in medical settings, where the omission of essential information can lead to inadequate clinical decisions or treatment recommendations [6,8]. Moreover, flawed, biased, or incomplete data can propagate inaccuracies and inconsistencies within the model outputs, reducing their reliability and safety, especially in healthcare applications [94,95]. Additionally, LLMs are limited by the cutoff date of their training data, causing reliance on outdated information, which diminishes the clinical relevance and accuracy of their responses [95,96]. These findings highlighted the interconnected nature of limitations in the LLMs, emphasizing that the core issues in output quality stem from deficiencies in the input data. Training data plays a crucial role, significantly impacting the quality and comprehensiveness of generated content. In this regard, our study indicated that while output limitations are numerous, their root causes lie primarily in the constraints and imperfections of the input data, underscoring the dependency of LLM outputs on input data quality.

The literature presents several strategies to improve the quality and comprehensiveness of content generated by LLMs. Some strategies emphasize enhancing the input quality to ultimately improve output quality. For example, role-playing prompts significantly increase accuracy, comprehensiveness, and acceptability of responses by guiding LLMs to adopt specific personas or roles, which encourages more detailed and contextually appropriate answers, as demonstrated in models like ChatGPT-4 and ChatGPT-3.5 [97]. Additionally, prompt engineering and augmentation—such as designing carefully structured prompts and incorporating relevant context or supporting data—enhance response quality. Retrieval augmented generation (RAG) techniques, which integrate verified external data into the generation process, further improve the accuracy and comprehensiveness of responses [98].

Other strategies focus on refining the training of LLMs to elevate the quality of generated content. Multi-modal and task-specific training, which involves training models on diverse data types like text combined with images and specialized datasets, allows LLMs to better manage complex healthcare questions with greater depth and clarity [99]. Fine-tuning and adaptation using domain-specific datasets, particularly healthcare data, improve the models' performance on specialized tasks such as medical question answering and clinical note summarization. Techniques like instruction tuning and rein-forcement learning from human feedback (RLHF) enhance model alignment with clinical expectations, thereby improving precision and relevance [3].

Some strategies center on improving feedback mechanisms for LLM outputs to enhance response quality. Systematic evaluation and benchmarking with healthcare-specific metrics—focusing on factual accuracy, medical reasoning, and readability (e.g., USMLE, PubMedQA)—support the continuous assessment and improvement of model outputs [99]. Moreover, incorporating human oversight through expert review and iterative feedback during model development and deployment promotes safety, ethical integrity, and clinical appropriateness of responses [97].

The study findings also identified an insufficiency in clinical nuance as a notable limitation of LLMs [17,20,23,30,35,37,45,53,57,58,60,66,68,72,73,76,78–82]. In this regard, the evidence indicates that, owing to ethical constraints and the highly specialized nature of the medical domain, the acquisition, processing, and utilization of clini-cal data by LLMs are subject to substantial restrictions. Furthermore, LLMs are not explicitly trained on medical records curated and selected by qualified clinical professionals [100]. Collectively, these factors likely constitute the primary under-lying causes of the limited clinical nuance observed in LLM outputs.

The literature outlines several strategies to address ethical issues in healthcare content generation by the LLMs. These strategies can be broadly categorized into governance, technical, input-focused, and output-focused approaches. Gover-nance strategies emphasize human oversight, including the establishment of legal and policy frameworks to ensure data governance and accountability [101,102]. Clinician-in-the-loop review and participatory development processes help main-tain human expertise and reduce overreliance on automated outputs [103]. Continuous real-world validation is essential for ensuring reliability, especially in diverse linguistic contexts [104]. Technical strategies focus on explainable AI methods and audit trails to foster trust and accountability [105]. Ethical considerations for model inputs include safety-first prompt-ing techniques designed to enhance ethical reasoning in AI outputs [106,107]. Finally, output-focused strategies advocate for standardized disclaimers that clarify AI limitations and help minimize risks of misinterpretation [108].

Our study findings indicated that LLMs face significant risks associated with the dissemination of harmful information or misinformation, particularly in less commonly used languages [32,67,76,77,80,89]. In this regard, evidence indicates that human evaluation methods are costly and difficult to standardize, especially for rapidly evolving LLMs [109]. On the other hand, novel approaches, enhanced with the optimized algorithm, have achieved greater accuracy, faster detection, and improved scalability than traditional methods, offering a robust and reliable defense against these threats [110,111]. Open-source benchmarks derived from physician certification exams—such as MedQA, PubMedQA, and the Massive Multitask Language Understanding (MMLU) dataset—are widely used to assess the knowledge and reasoning of medical LLMs, including those promoted for patient-facing use [112–115]. In such context, benchmark-based evaluations can serve as a scalable proxy for identifying model behaviors that could adversely affect patient care.

Despite recent advancements in LLMs across various aspects of input, processing, and output capabilities, our study provides valuable insights into the limitations of LLMs in delivering healthcare content [116–118]. These insights can be leveraged by relevant stakeholders to develop precise, evidence-based strategies aimed at mitigating these limitations while preserving the strengths of LLMs. While newer versions of LLMs continue to be released, they still exhibit certain limitations [117,119]. Consequently, the data presented in our study remains highly relevant and valuable to various stakeholders, including LLM developers, healthcare policymakers, and administrators aiming to implement LLM technologies within their organizations.

## 4. Limitations and implications

This study had a notable limitation in that it did not include data from studies published in languages other than English due to time and resource constraints. Such a limitation may have resulted in the exclusion of relevant existing data on the topic published in other languages. Consequently, future researchers are encouraged to conduct similar reviews that encompass non-English publications. The study also did not evaluate heterogeneity across different study types, nor did it distinctly delineate limitations specific to clinical, educational, and administrative contexts. On the other hand, this study offers several important implications for key stakeholders, including healthcare policymakers, administrators, researchers, and developers of large language models. Specifically, the findings highlighted output-related limitations as the most significant category of constraints affecting LLMs in the provision of healthcare content. In this regard, the most frequently cited limitation was the accuracy gap. This identification of the areas with the greatest vulnerability and weakness can guide beneficiaries in prioritizing and implementing strategic and operational plans, while also informing researchers in their efforts to mitigate these limitations in future developments. Meanwhile, our study findings indicated that although limitations in LLM outputs are numerous, their primary root causes lie in the constraints and imperfections of the input data. This underscores the critical dependency of LLM output quality on the integrity and quality of the input data. This finding also has significant implications for the beneficiaries. In this regard, our study proposed several strategies to address major issues encountered by the LLMs in the provision of healthcare content. These solutions are designed to be practical and can also be readily adopted by beneficiaries to improve healthcare outcomes and ensure safer, more reliable LLM-assisted content generation.

## 5. Conclusion

The study identified several categories of limitations in LLMs regarding the provision of healthcare content, with output limitations being the most prevalent. In this regard, the most frequently cited limitation was the accuracy gap (i.e., fabricated responses). Our findings indicated that these numerous output limitations primarily stem from constraints and imperfections in the input data. This highlights the critical dependency of LLM output quality on input data integrity. Additionally, our study proposed several strategies to address these key challenges encountered by the LLMs.

## 6. Methods

This study was a systematic review conducted in 2025, following the guidelines set forth by the Preferred Reporting Items for Systematic Reviews and Meta-Analyses (PRISMA) statement to ensure transparent and standardized reporting of its methodology and findings [120]. Following the completion of the systematic review, a thematic analysis was performed on the collected data. The research question was formulated using the PICOT framework, which includes Population (P), Intervention or Indicator (I), Comparison group (C), Outcome of interest (O), and Timeframe (T) [121]. In this study, the population was defined as the existing LLMs operating globally; the intervention as the limitations of LLMs in provision of healthcare content, the comparison group encompassed the various types of LLMs reported in the literature, the outcome focused on limitations occurring in the input, process, and output phases, and the timeframe was set from 2018 to 2025. The research question was formulated as follows: "What are the limitations of large language models in the generation of healthcare content in the 2018-2025 time-period?"

### 6.1. Information sources and search strategy

A comprehensive literature search was conducted in September 13, 2025, to identify all articles published in English between 2018 and 2025 that address the limitations of large language models in generating healthcare content. The specified time span was chosen because the initial release of LLMs dates back to 2018 [122]. The databases searched included PubMed, Scopus, and Cochrane database of systematic reviews. MeSH terms were used to categorize keywords into three groups: limitations, large language models, and healthcare. Synonymous keywords within each group were combined using the logical operator "OR," and the three groups were subsequently combined using the logical operator "AND." The search was performed across titles, abstracts, and keywords. Reference management was performed using EndNote version 20.2.1. The detailed search strategy is presented in Table 2.

### 6.2. Study selection

Following the database search, duplicate articles were removed, and the remaining records were screened based on their titles and abstracts. Articles that were not aligned with the research objectives were excluded, and the full texts of the eligible articles were thoroughly reviewed. Only those meeting the predefined inclusion criteria were incorporated into the final analysis. This entire screening and selection process was independently conducted by two authors. In cases of disagreement, a third author was consulted to resolve discrepancies and finalize the screening process.

  Inclusion criteria:

- Availability of full text.

- Published in English language.

- Published after 2018.

  Exclusion criteria:

- Published solely in conferences. This consideration arose from the perception that the quality of peer review for conference papers is generally lower compared to manuscripts published in journals indexed in prominent databases, which undergo a more rigorous and extensive peer review process prior to publication. In this context, the authors concluded that including conference papers could introduce a degree of bias and compromise the overall quality of the study data.

- Published as letter to editor or protocol. The exclusion was based on the insufficient or lower quality of data presented in these types of research papers. This approach was adopted to ensure the higher quality of data reported in the current study.

### 6.3. Data quality assessment

The quality of all included studies was independently assessed by two evaluators using the Authority, Accuracy, Coverage, Objectivity, Date, and Significance (AACODS) checklist, which consists of six items [123]. The checklist was selected due to its straightforwardness and clarity, as it effectively presents the quality of the included studies in a transparent and easily understandable manner. A standardized scoring system was applied, assigning 2 points for "Yes," 1 point for "Can't Tell," and 0 points for "No," resulting in scores ranging from 0 to 12, with higher scores indicating better quality. Studies were classified into four categories based on their scores: very low quality (0–3), low quality [4–6], medium quality [7–9], and high quality [10–12]. In the process, only studies rated as medium or high quality were considered eligible for inclusion in the research. Any disagreements between the two evaluators were resolved through discussion and consultation with a third reviewer acting as an arbitrator. To ensure consistency, this evaluation procedure was conducted twice.

**Table 2. Search strategy.**

| Final Strategy | "Limitations" AND "Large language models" AND "Healthcare" |
|---|---|
| **Context** | **Keywords** |
| Limitations | limitation* OR restriction* OR constraint* OR boundar* OR barrier* OR challenge* OR impediment* OR drawback* OR weakness* OR deficienc* OR defect* |
| Large Language Models | large language model* OR LLM OR LLMs OR ChatGPT OR Bard OR Claude OR Copilot OR Bing OR Gemini OR Perplexity OR ChatSonic |
| Healthcare | healthcare OR health |
| **Database-specific search strategies** | |
| **Database** | **Search strategy** |
| PubMed | ((limitation*[Title/Abstract] OR restriction*[Title/Abstract] OR constraint*[Title/Abstract] OR boundar*[Title/Abstract] OR barrier*[Title/Abstract] OR challenge*[Title/Abstract] OR impediment*[Title/Abstract] OR drawback*[Title/Abstract] OR weakness*[Title/Abstract] OR deficienc*[Title/Abstract] OR defect*[Title/Abstract]) AND (Large Language Models [MeSH Terms] OR large language model*[Title/Abstract] OR LLM[Title/Abstract] OR LLMs[Title/Abstract] OR ChatGPT[Title/Abstract] OR Bard[Title/Abstract] OR Claude[Title/Abstract] OR Copilot[Title/Abstract] OR Bing[Title/Abstract] OR Gemini[Title/Abstract] OR Perplexity[Title/Abstract] OR ChatSonic[Title/Abstract])) AND (Health [MeSH Terms] OR healthcare[Title/Abstract] OR health[Title/Abstract]) |
| Scopus | (TITLE-ABS-KEY (limitation* OR restriction* OR constraint* OR boundar* OR barrier* OR challenge* OR impediment* OR drawback* OR weakness* OR deficienc* OR defect*) AND TITLE-ABS-KEY (large language model* OR LLM OR LLMs OR ChatGPT OR Bard OR Claude OR Copilot OR Bing OR Gemini OR Perplexity OR ChatSonic) AND TITLE-ABS-KEY (healthcare OR health)) |
| Cochrane database of systematic reviews | limitation* OR restriction* OR constraint* OR boundar* OR barrier* OR challenge* OR impediment* OR drawback* OR weakness* OR deficienc* OR defect* in Title Abstract Keyword AND large language model* OR LLM OR LLMs OR ChatGPT OR Bard OR Claude OR Copilot OR Bing OR Gemini OR Perplexity OR ChatSonic in Title Abstract Keyword AND healthcare OR health in Title Abstract Keyword (Word variations have been searched) |

## 6.4. Data extraction

Data extraction from the selected articles was performed independently by two authors. A third author supervised and approved the data extraction process. A data extraction form was developed using Microsoft Excel 2016, which included sections for the study citation, year of publication, and a summary of the findings.

## 6.5. Data analysis

The data collected from the preceding steps were analyzed using the qualitative thematic approach proposed by Boyatzis, employing an inductive methodology. This approach comprises multiple steps, including familiarization with the study data, generation of initial codes, development of themes, and finally reviewing, defining, and naming these themes [124]. The thematic analysis was also performed using the input–process–output (IPO) model as the foundational framework. The IPO model, also referred to as the input-process-output pattern, is a fundamental framework extensively utilized in systems analysis and software engineering. It serves as a foundational approach for representing the structure of an information processing system or other procedural workflows. This model is commonly introduced in introductory programming and systems analysis literature as the most basic and essential method for describing and conceptualizing a process [125–128].

The term input refers to the resources, data, or materials introduced into a system. The process encompasses the operations or transformations applied to the input to convert it into an output. And, the output denotes the results or products generated following the processing of the input [129]. In such context, in our study, the category labeled input was defined to include a collection of themes related to the resources, data, or materials introduced into LLMs. The process category was assigned to themes corresponding to the operations or transformations applied to the input provided to the LLMs, facilitating its conversion into an output. Similarly, the output category was designated for themes associated with the results or products generated as a consequence of processing the inputs given to the LLMs.

The authors systematically reviewed all the extracted data from the included studies to gain a comprehensive understanding, assigning initial codes to each significant data segment. Each code was developed to capture a unique outcome of artificial intelligence in healthcare. Subsequently, codes sharing similar concepts were grouped into sub-themes, and related sub-themes were further consolidated under overarching main themes. Prior to this categorization, all initial codes underwent thorough examination and refinement. The finalized sub-themes and main themes were then defined, described, and presented in a tabular format. Any disagreements among the authors were resolved through mutual consultation.

To ensure the validity and reliability of the qualitative data analysis, the authors adhered to Lincoln and Guba's criteria, which encompass credibility, transferability, dependability, and confirmability [130]. In this regard, credibility was established through repeated cross-referencing of the codes with their original data sources. Dependability was ensured by having two authors independently conduct the thematic analysis and compare results to identify discrepancies. Confirmability was maintained via mutual review of the themes, sub-themes, and codes by the authors. Finally, transferability was enhanced by expressing the findings in a manner that facilitates application across diverse contexts within the healthcare system.

## Supporting information

**S1 Appendix. Quality assessment.**
(DOCX)

**S1 Checklist. PRISMA checklist [120].**
(DOCX)

## Acknowledgments

The authors utilized ChatGPT-4 in order to rewrite the entire text of the manuscript in terms of correct grammar and wording. In this regarding, the authors validated the clarity and accuracy of the rewritten text.

## Author contributions

**Conceptualization:** Mohsen Khosravi.

**Data curation:** Mohsen Khosravi, Seyyed Morteza Mojtabaeian.

**Formal analysis:** Mohsen Khosravi, Zahra Zamaninasab.

**Investigation:** Mohsen Khosravi, Seyyed Morteza Mojtabaeian.

**Methodology:** Mohsen Khosravi.

**Project administration:** Mohsen Khosravi.

**Resources:** Seyyed Morteza Mojtabaeian.

**Validation:** Emine Kübra Dindar Demiray, Morteza Arab-Zozani.

**Visualization:** Zahra Zamaninasab.

**Writing – original draft:** Mohsen Khosravi.

**Writing – review & editing:** Zahra Zamaninasab, Emine Kübra Dindar Demiray, Morteza Arab-Zozani.

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
