## [Decision Letter · Decision Letter 0]

28 Jan 2026

Response to Reviewers'. This file does not need to include responses to any formatting updates and technical items listed in the 'Journal Requirements' section below.'. This file does not need to include responses to any formatting updates and technical items listed in the 'Journal Requirements' section below.* A marked-up copy of your manuscript that highlights changes made to the original version. You should upload this as a separate file labeled 'Revised Manuscript with Track Changes'.'.* An unmarked version of your revised paper without tracked changes. You should upload this as a separate file labeled 'Manuscript'.'. If you would like to make changes to your financial disclosure, competing interests statement, or data availability statement, please make these updates within the submission form at the time of resubmission. Guidelines for resubmitting your figure files are available below the reviewer comments at the end of this letter. We look forward to receiving your revised manuscript. Kind regards, Mayue Shi, Ph.D.Academic EditorPLOS Digital Health Leo Anthony CeliEditor-in-ChiefPLOS Digital Healthorcid.org/0000-0001-6712-6626 **Journal Requirements:** If the reviewer comments include a recommendation to cite specific previously published works, please review and evaluate these publications to determine whether they are relevant and should be cited. There is no requirement to cite these works unless the editor has indicated otherwise.  **Additional Editor Comments (if provided):** This is a timely systematic review that addresses an important and rapidly evolving topic in healthcare AI. The review is comprehensive and well motivated; however, several key issues should be addressed to strengthen its rigor and clarity. I suggest authors addressing reviewers' concerns accrodingly. Methodological transparency could be improved by providing reproducible, database-specific search strategies, clarifying quality and bias assessment methods, and importantly, justifying exclusion of conference literature. The discussion would benefit from deeper analytical integration.**Reviewers' Comments:** Reviewer's Responses to Questions

**Comments to the Author**

1. Does this manuscript meet PLOS Digital Health’s publication criteria? Is the manuscript technically sound, and do the data support the conclusions? The manuscript must describe methodologically and ethically rigorous research with conclusions that are appropriately drawn based on the data presented.? Is the manuscript technically sound, and do the data support the conclusions? The manuscript must describe methodologically and ethically rigorous research with conclusions that are appropriately drawn based on the data presented.

Reviewer #1: Yes

Reviewer #2: Yes

Reviewer #3: Yes

2. Has the statistical analysis been performed appropriately and rigorously?

Reviewer #1: Yes

Reviewer #2: N/A

Reviewer #3: Yes

3. Have the authors made all data underlying the findings in their manuscript fully available (please refer to the Data Availability Statement at the start of the manuscript PDF file)?

The PLOS Data policy requires authors to make all data underlying the findings described in their manuscript fully available without restriction, with rare exception. The data should be provided as part of the manuscript or its supporting information, or deposited to a public repository. For example, in addition to summary statistics, the data points behind means, medians and variance measures should be available. If there are restrictions on publicly sharing data—e.g. participant privacy or use of data from a third party—those must be specified.requires authors to make all data underlying the findings described in their manuscript fully available without restriction, with rare exception. The data should be provided as part of the manuscript or its supporting information, or deposited to a public repository. For example, in addition to summary statistics, the data points behind means, medians and variance measures should be available. If there are restrictions on publicly sharing data—e.g. participant privacy or use of data from a third party—those must be specified.

Reviewer #1: Yes

Reviewer #2: Yes

Reviewer #3: Yes

4. Is the manuscript presented in an intelligible fashion and written in standard English?

Reviewer #1: Yes

Reviewer #2: Yes

Reviewer #3: Yes

Reviewer #1: 1. This study only includes articles published in English. This is a significant limitation for a systematic review because valuable research in other languages, such as Chinese, German, and Arabic, is important in artificial intelligence and healthcare. Excluding these sources may skew our understanding of global issues, especially regarding translation quality. It is crucial to reassess this limitation or at least discuss its effects clearly.

2. The Discussion section primarily repeats the results without fully exploring the connections between the input, process, and output limitations. A more in-depth analysis is needed to understand how input limitations, like a lack of Arabic datasets, lead to specific output problems, such as variable translation quality.

3. In the Conclusion and Limitations section (Section 4), it is important to state clearly that reliance on English publications is a major limitation. This focus could distort the findings and underreport language-specific biases. Future research should address this issue by including studies in other languages to improve the generalizability of the findings.

4. The authors should better emphasize the connection between input flaws and output problems. The text states that output issues "mainly resulted from flaws in input data," but it should elaborate on this point. For example, one could state, "Limited multimodal data processing (input flaw) leads to incomplete responses and neglected imaging descriptions (output limitation) in radiology contexts. "A Hybrid Autoencoder and Gated Recurrent Unit Model Optimized by Honey Badger Algorithm for Enhanced Cyber Threat Detection in IoT Networks”. This reference discusses a hybrid deep learning model in the context of IoT security and relates directly to the limitations in the review of large language models (LLMs). Specifically:

5. It shows the need for specialized hybrid architectures and optimization algorithms, in contrast to the general-purpose LLMs discussed. This reference highlights a successful application in a specific field (cyber threat detection), presenting a counterexample to the weaknesses of general LLMs in complex fields such as healthcare.

6. The need to optimize parameters reflects the LLM issue regarding reliance on general training, which leads to gaps in accuracy and biases.This reference should be included in the Discussion section while analyzing the relationship between output limitations and flaws in model design.

7. The text notes that large language models (LLMs) are prone to generating plausible but incorrect or fabricated information, a phenomenon known as It is important to emphasize this term in bold for clarity.

8. In Section 6.5, it is helpful to reference the original work that defines the IPO model to maintain academic rigor. Although the model is introduced, clarifying its ties to system analysis will improve understanding. It is suggested that the authors check the alignments in the reference section and front issues and correct anything if needed.

Reviewer #2: Summary

This manuscript presents a systematic review (2018–2025) that synthesizes the limitations of large language models (LLMs) in generating healthcare content, using the input–process–output (IPO) framework. The authors searched PubMed, Scopus, and Cochrane, identified 81 studies, assessed them using the AACODS checklist, and performed an inductive thematic analysis following Boyatzis’s method. The study identifies eight themes relating to data limitations, prompt/input dependence, architectural constraints, interaction challenges, output quality limitations, and ethical/regulatory risks. The authors conclude that although output errors are the most commonly cited limitations, many originate from input-quality constraints, and they propose strategies to mitigate these issues.

Major Comments

The search strategy is overly minimal (“Limitations” AND “Large language models” AND “Healthcare”), and Table 1 lists broad synonym groups without showing the actual Boolean-structured queries used in each database. For reproducibility, PRISMA requires database-specific queries (e.g., PubMed MeSH lines).

There is no mention of PROSPERO registration. Given the systematic review methodology, the absence of protocol registration raises concerns about potential bias and post-hoc modifications.

The manuscript excludes conference papers and letters but does not justify these exclusions. Many influential AI/LLM limitations papers appear in conference proceedings (ACL, NeurIPS, AAAI).

While the IPO model is a general systems-analysis tool, its adoption here seems partially post-hoc; some limitations overlap categories (e.g., prompt issues can be both input and process). Consider clarifying the rationale for mapping each limitation to its category.

The results extend to many pages with repetitive descriptions of similar limitations. While comprehensive, it compromises readability. A more synthesized narrative plus thematic tables would improve clarity.

The figure 2 displays frequency counts but lacks:

definition of how “number of references” was computed (e.g., per theme or per sub-theme),

visual clarity (font size, grouping).

Much of the discussion repeats results rather than integrating them with broader literature or offering conceptual insight. The section could more deeply analyze why certain limitations dominate and how emerging LLM architectures (e.g., multimodal models, RAG, domain-specific pretraining) may mitigate them.

Recent advances such as GPT-5, Gemini 2.0, Qwen 2.5, smaller <13B models like Gemma and domain-specific models (Med-PaLM, MedGemma, LLaMA-family medical finetunes) are not discussed, though their capabilities meaningfully impact the identified limitations.

The study acknowledges language restriction but omits several important limitations:

no assessment of publication bias,heterogeneity across study types (chatbot assessments, narrative essays, technical papers), and limited granularity in separating limitations specific to clinical, educational, and administrative contexts

Numerous grammatical inconsistencies remain (e.g., “LLMS” instead of LLMs; “supposedly to be derived…”).\

The abstract has several duplicated or redundant phrases.

Improve flow by reducing overly long sentences throughout the manuscript.

Ensure consistent referencing formatting (several citations appear as ranges with missing formatting).

Citation [75] is listed as “!!! INVALID CITATION !!!” which must be corrected.

75. . !!! INVALID CITATION !!! [4, 5, 12, 24, 29, 30, 41, 43, 51, 57, 59, 62, 67-70].

Table 2 is overly long and may be better placed in an appendix.

Some themes overlap significantly (e.g., “dependence on input quality” vs “dependence on prompt quality”). Consider merging or refining boundaries.

Terminology such as “accuracy gap” should be defined more precisely.

The authors mention using ChatGPT-4 to “rewrite the text,” which is appropriate, but PLOS typically requires specifying:

which sections were rewritten,

whether human authors verified accuracy.

Reviewer #3: This is a great work. It is timely and important, and it has a lot of present and future implications on both healthcare practice and research. The coverage and classification of the LLM limitations and the awareness about them are essential for different backgrounds and academic levels of people involved in designing, implementing and end-using them. Going forward, this subject can not be avoided and no matter what the limitations and challenges are, we have to face them, deal with them and tame them. Thanks for sharing this important work and thanks for giving me the opportunity to review it.

I learned a lot from reading it. The researchers have summarized many ideas from the literature that they have reviewed, that I would not have time to read it myself and add to my understanding. So I have to be extremely thankful.

One of the injustices that afflict this study is that it is tackling a huge subject. While reading it, at some points or sub-topics, I found myself thinking that every one limitation or challenge mentioned here is worth a separate study because of the depth of the ideas discussed.

I would like to put my comments in the following bullets, following quoting the manuscript text portions hoping that this will help in shaping the final publication in the best shape.

• “Two independent evaluators screened the references and assessed quality of the selected studies using the Authority, Accuracy, Coverage, Objectivity, Date, and Significance (AACODS) checklist, which examines Authority, Accuracy, Coverage, Objectivity, Date, and Significance.” In this sentence, the evaluators assessed the Authority, Accuracy etc of what exactly? For readers with different backgrounds, the second subject of this sentence needs to be clearer.

• “Boyatzis's qualitative thematic approach” again, for any reader, they would expect some elaboration on this approach anywhere in the manuscript. The approach has been mentioned four times, including the one in the references without at least a single line simple explanation about it.

• “A total of 81 studies were included in the final analysis. The included studies were predominantly of high quality and demonstrated minimal risk of bias.” Looking for the methodology used to assess risk of bias.

• “The thematic analysis identified key themes: data limitations, dependence on input and prompt quality, accessibility issues, model design and architecture constraints, interaction challenges, response quality and comprehensiveness, and ethical, safety, and regulatory concerns.” Tables? I would suggest that multiple tables should replace the huge Table 2, to ensure that the text is shown in correspondence with the relevant table contents.

• “The study identified multiple limitations of LLMs in healthcare, with output issues being most common.” Reading through the text and the table, I found it necessary to repeat the point of dividing Table 2 into more tables, and this might bring even more subtopics that deserve to be included in the tables.?

• “However, these output problems mainly resulted from flaws in input data, emphasizing the crucial 60 role of input quality. The study also proposed strategies to address these challenges.” Table?

• “The search yielded 81 studies, and the quality of the included studies was presented to be predominantly high.” Quality assessment criteria?

• “The thematic analysis yielded a number of themes and sub-themes.” Tables?

• “The results of the research found that while the main area of LLMs` limitations is corresponding to the outputs, and particularly the existing gaps in accuracy, these limitations are supposedly to be derived from the existing flaws in the input data.” Important conclusion to be emphasized in the discussion.

• “The study also presented some strategies to overcome these limitations 80 based on the existing data within the literature.” Important future plan or recommendation.

• “Large language models (LLMs) are sophisticated artificial intelligence (AI) systems developed through extensive training on vast corpora of text data, enabling them to generate outputs that closely resemble human language(1). These models have been widely utilized across diverse medical domains, including health informatics, medical imaging, clinical diagnostics, treatment planning, ophthalmology, oncology, and other specialized fields(2). This trend signifies their broad and growing integration into medical research and clinical practice. LLMs have become pivotal in healthcare by enhancing clinical decision support, diagnostics, medical education, and patient engagement(2-4). They improve diagnostic accuracy by analyzing extensive clinical data and medical literature, aiding in personalized treatment planning and patient care management(3).” Debatable, but understood as completely dependant on the reference.

• “98 Notwithstanding the numerous advantages previously discussed, LLMs in healthcare exhibit

99 several critical limitations that must be addressed to ensure their safe and effective deployment.

100 In this regard, they are prone to generating plausible yet factually incorrect or fabricated

101 information, a phenomenon known as hallucination, which poses significant risks in clinical

102 settings(3, 6).” Table for limitations, hallucinations included.

• “Furthermore, LLMs often lack the depth of contextual understanding required to

103 accurately interpret complex medical scenarios, as they may fail to integrate multifaceted clinical

104 data or temporal information adequately(3).” Other limitations, lack of depth, lack of contextuality, failure to integrate clinical data, failure to integrate temporal information. Table?

• “The development and application of LLMs are also

105 constrained by limited access to high-quality, diverse clinical datasets due to privacy, ethical, and

106 legal challenges(7). Ethical and legal issues such as bias, misinformation, data privacy, and

107 insufficient regulatory frameworks further challenge their adoption(8)” Constraints to LLM development and constraints to LLM adoption.

• “Additionally, the opaque,

108 "black box" nature of these models undermines transparency and interpretability, complicating

109 trust and reliance by healthcare professionals(9).” Limitation table: opacity, black box, trust and reliance.

• “Practical concerns include their substantial

110 computational and energy demands, which limit feasibility in resource-constrained

111 environments(3).” Practical limitations, table?

• “Such limitations of LLMs pose significant challenges to their adoption and

112 utilization across various sectors of healthcare systems. In this regard, these challenges must be

113 critically addressed to enable the successful and comprehensive integration of LLM technologies

114 within healthcare infrastructure.” Consequences of limitations. Table or figure.

• “115 While several review studies have aimed to delineate the limitations of LLMs in providing

116 healthcare content, there remains a significant gap in the literature regarding a comprehensive

117 and systematic presentation of these limitations for end-users” Practical limitations in LLMs, research gap in LLM limitations.

• “In this regard, a systematic

118 review categorized the limitations of LLMs into two primary domains: design and output.” In this sentence, “ a systematic review”, should not it be “this systematic review”?

• “Design

119 limitations included several items such as lack of optimization for the medical domain, data

120 transparency issues, and accessibility challenges.” Table?

• “Output limitations also included several items

121 such as non-reproducibility, incompleteness, inaccuracies, safety concerns, and biases(6).” Table?

• “The

122 data generated from such research would be invaluable for technology developers to enhance

123 the quality of their models.” In the introduction, “such research”? Again, are we still referring to this research or recommending such research in our introduction?

• “Additionally, healthcare policymakers and administrators could

124 leverage this information to make informed decisions about implementing these technologies

125 within their organizations, fully acknowledging their current constraints.” Definitely

• “Moreover, future

126 researchers would benefit from this detailed framework by conducting focused investigations on

127 each identified limitation, thereby contributing further insights and advancing the field for

128 subsequent users and stakeholders.” I totally agree.

• “” Why is the methods or material section put after the conclusion? Little mentioned about the methodology of the research under the ‘Results’. Figure-1

• Figure 1 “Records removed before screening: Duplicate records removed (n =1798) Records marked as ineligible by automation tools (n =0) Records removed for other reasons (n =0)” Could you please elaborate on the automation tools? What are the other reasons?

• “Figure 1. PRISMA diagram.” Would this be explained to give the lacking ‘Methods’ section?

• “In this regard, 20% of the included studies were published in 2023, 139 40% in 2024, and the remaining 40% in 2025.” “In this regard” this phrase was overused in this script. It could be replaced by different places in different places. What is the significance of the distribution of the studies onto the years? Can this be added to the research questions and given explanation for its significance?

• “2.1. Data quality 181 The quality assessment of the included studies indicated that they were predominantly of high 182 quality, with an average score of 10. Approximately 34% of the studies achieved the highest 183 quality score of 12, while about 8% scored 7, reflecting lower quality relative to the other 184 included studies. Moreover, the level of bias in the included studies was generally low, with 185 approximately 59% of the studies exhibiting the minimal possible bias (Appendix 1. Quality 186 Assessment).” The assessments of both quality and bias need to be explained in the text, in addition to any table. Also, the table in Appendix one is very brief. It shows the questions for the tabulated criteria, but I believe it needs more elaboration. The evaluation is generally subjective in nature.

• “There was also an identified over-reliance on imaging modalities such as

254 CT and MRI, without adequate customization for specific clinical contexts” This is very important point. I believe it deserves wider elaboration, but I understand that it might over expand the scope of this work.

• “255 Furthermore, LLMs like ChatGPT demonstrated an inability for critical thinking necessary to tailor

256 and guide patient management effectively” I believe that for an LLM to have critical thinking, they would make them more challenging than any challenge or limitation mentioned here. However, guiding their thinking by human supervised training, and training and more training would make them more useful.

• “Conversation tracking posed

265 further limitations: ChatGPT 3.5 lacked conversation memory due to privacy and browsing

266 restrictions, while ChatGPT 4.0, despite tracking conversations, inaccurately counted inquiries,

267 thereby undermining feedback and dialogue continuity” Conversation memory, and tracking accuracy would be one of the biggest future solutions for many of the challenges in LLM clinical use.

• “. Accuracy

293 gaps were evident, with occasional imprecision, indecisiveness, hallucinations, irrelevant

294 information, and omissions of key clinical considerations, particularly for special populations

295 such as pregnant patients; fabricated references were also observed sporadically” I realize that fabricated references are a feature of inaccuracy, but I would still suggest that this might better be a separate sentence.

• “2.2.3.2. Ethical, safety, and regulatory concerns” This is one of the richest sections of the manuscript. I believe it deserves a separate table or tables. Focusing on keywords and their explanations. It would help a lot in this research to coin new definitions for many keywords to standardize the future study and tackling of the LLMs limitations and challenges

• “Table 2. Thematic analysis of findings.” A repeated comment on this table is that it can have its text content converted to single keywords and this can even help create figure and charts from it.

• “In this regard, in line with our study findings, a recent review emphasized 347 that the taxonomy of limitations associated with large language models reveals a substantially 348 greater number of codes related to output issues compared to those connected with design or 349 input phases.” Again, “in this regard”, I would suggest that the researchers could take the opportunity to claim the taxonomy of LLM limitations and standardize their linguistic use.

• “Figure 2. Distributions of limitations of LLMs, categorized according to the frequency of citations reported in the literature.” Again, I would recommend dividing this figure into three figures. It looks very crowded, and the text font is very tiny because of that.

• “381 8). Moreover, flawed, biased, or incomplete data can propagate inaccuracies and inconsistencies

382 within the model outputs, reducing their reliability and safety, especially in healthcare

383 applications” To explain this again. I give the researchers the richness of their work, but the bias and quality standards need to be explained so the reader can benefit from the manuscript and build upon it in improving their future clinical interactions with the LLMs

• “. This finding also has significant

445 implicationsfor the beneficiaries. In this regard, our study proposed several strategies to address

446 major issues encountered by the LLMs in the provision of healthcare content.”

Thanks again,

Yasser Abdullah

**Do you want your identity to be public for this peer review?** For information about this choice, including consent withdrawal, please see our Privacy Policy..

Reviewer #1: No

Reviewer #2: **Yes:** Balu BhasuranBalu BhasuranBalu BhasuranBalu Bhasuran

Reviewer #3: **Yes:** Yasser AbdullahYasser AbdullahYasser AbdullahYasser Abdullah

**Figure resubmission:**  While revising your submission, we strongly recommend that you use PLOS’s NAAS tool (https://ngplosjournals.pagemajik.ai/artanalysis) to test your figure files. NAAS can convert your figure files to the TIFF file type and meet basic requirements (such as print size, resolution), or provide you with a report on issues that do not meet our requirements and that NAAS cannot fix. 

**Reproducibility:** To enhance the reproducibility of your results, we recommend that authors of applicable studies deposit laboratory protocols in protocols.io, where a protocol can be assigned its own identifier (DOI) such that it can be cited independently in the future. Additionally, PLOS ONE offers an option to publish peer-reviewed clinical study protocols. Read more information on sharing protocols at https://plos.org/protocols?utm_medium=editorial-email&utm_source=authorletters&utm_campaign=protocols To enhance the reproducibility of your results, we recommend that authors of applicable studies deposit laboratory protocols in protocols.io, where a protocol can be assigned its own identifier (DOI) such that it can be cited independently in the future. Additionally, PLOS ONE offers an option to publish peer-reviewed clinical study protocols. Read more information on sharing protocols at https://plos.org/protocols?utm_medium=editorial-email&utm_source=authorletters&utm_campaign=protocols

---

## [Decision Letter · Decision Letter 1]

21 Mar 2026

A Systematic Review of the Limitations of Large Language Models in Generating Healthcare Content

PDIG-D-25-00961R1

Dear Dr. Khosravi,

We are pleased to inform you that your manuscript 'A Systematic Review of the Limitations of Large Language Models in Generating Healthcare Content' has been provisionally accepted for publication in PLOS Digital Health.

Best regards,

Zhenwei Shi

Section Editor

PLOS Digital Health

**Additional Editor Comments (if provided):**

After considering the reviewers’ comments, I am pleased to inform you that your manuscript has been accepted for publication. I have no further comments at this stage. Please follow the instructions regarding the next steps in the publication process.

Congratulations, and thank you for submitting your work to us.

**Reviewer Comments (if any, and for reference):**

Reviewer's Responses to Questions

**Comments to the Author**

Reviewer #1: All comments have been addressed

Reviewer #2: All comments have been addressed

Reviewer #3: All comments have been addressed

publication criteria? Is the manuscript technically sound, and do the data support the conclusions? The manuscript must describe methodologically and ethically rigorous research with conclusions that are appropriately drawn based on the data presented.? Is the manuscript technically sound, and do the data support the conclusions? The manuscript must describe methodologically and ethically rigorous research with conclusions that are appropriately drawn based on the data presented.

Reviewer #1: Yes

Reviewer #2: Yes

Reviewer #3: Yes

3. Has the statistical analysis been performed appropriately and rigorously?

Reviewer #1: Yes

Reviewer #2: N/A

Reviewer #3: N/A

4. Have the authors made all data underlying the findings in their manuscript fully available (please refer to the Data Availability Statement at the start of the manuscript PDF file)?

The PLOS Data policy requires authors to make all data underlying the findings described in their manuscript fully available without restriction, with rare exception. The data should be provided as part of the manuscript or its supporting information, or deposited to a public repository. For example, in addition to summary statistics, the data points behind means, medians and variance measures should be available. If there are restrictions on publicly sharing data—e.g. participant privacy or use of data from a third party—those must be specified.requires authors to make all data underlying the findings described in their manuscript fully available without restriction, with rare exception. The data should be provided as part of the manuscript or its supporting information, or deposited to a public repository. For example, in addition to summary statistics, the data points behind means, medians and variance measures should be available. If there are restrictions on publicly sharing data—e.g. participant privacy or use of data from a third party—those must be specified.

Reviewer #1: Yes

Reviewer #2: Yes

Reviewer #3: Yes

5. Is the manuscript presented in an intelligible fashion and written in standard English?

Reviewer #1: Yes

Reviewer #2: Yes

Reviewer #3: Yes

Reviewer #1: The authors have done a thorough job revising the manuscript after the initial peer review. They have effectively addressed all the concerns and suggestions raised by the reviewers. The current version of the manuscript is much stronger and now meets the publication standards.

1. The authors thoroughly addressed reviewer comments, showing their dedication to enhancing the study’s methods and discussion based on feedback.

2. The manuscript is now clearer and more organized, with added search strategies and better definitions of analytical frameworks.

3. The technical quality is confirmed with criteria for study inclusion/exclusion and transparent bias and quality assessments.

4. This work significantly contributes to digital health by detailing the limitations of large language models in healthcare content.

5. Results are strengthened by better data organization, with tables and figures illustrating model constraint frequencies and distributions.

6. The manuscript's presentation and quality have greatly improved, with smoother text, corrected grammar, and recent references.

Reviewer #2: The authors have adequately addressed all of the points raised in the previous round of review. The manuscript is significantly improved, and I have no further concerns. I recommend the paper for publication.

Reviewer #3: I repeat my appreciation for the opportunity to review this work. I find it a great work and I believe it is timely and important to be added to the literature discussing the Healthcare LLM subject. Despite the fact that as a reader, I do not agree that one huge table can improve readability, I would recommend this manuscript to be published and I wish you the best of luck in your future research and I encourage you to build on this work in particular.

**Do you want your identity to be public for this peer review?** For information about this choice, including consent withdrawal, please see our Privacy Policy..

Reviewer #1: No

Reviewer #2: **Yes:** Balu BhasuranBalu BhasuranBalu BhasuranBalu Bhasuran

Reviewer #3: **Yes:** Yasser AbdullahYasser AbdullahYasser AbdullahYasser Abdullah
